# Antiproliferative Activity of *Stokesia laevis* Ethanolic Extract in Combination with Several Food-Related Bioactive Compounds; In Vitro (Caco-2) and In Silico Docking (TNKS1 and TNKS2) Studies

**Georgeta Neagu [1], Amalia Stefaniu [1], Adrian Albulescu [1,2], Lucia Pintilie [1] and Lucia Camelia Pirvu [1,*]**

[1] National Institute for Chemical-Pharmaceutical Research and Development, ICCF, 112 Vitan Av., 031299 Bucharest, Romania; getabios@yahoo.com (G.N.); astefaniu@gmail.com (A.S.); rockady2@gmail.com (A.A.); lucia.pintilie@gmail.com (L.P.)

[2] "Stefan S. Nicolau" Institute of Virology, 285 Mihai Bravu Av., 030304 Bucharest, Romania

* Correspondence: lucia.pirvu@yahoo.com

**Featured Application: Developing new natural or combined drugs with antitumor activity.**

**Abstract:** This study evaluates in vitro cytotoxic and antiproliferative activity on human colon tumor cell line Caco-2 (ATCC-HTB-37) of a standardized (5 mg GAE/mL) ethanolic extract from *Stokesia laevis* (Slae26), of five polyphenols compounds (reference substances, ref.), namely luteolin-7-O-glucoside, luteolin-8-C-glucoside, caffeic acid, gentisic acid, and p-aminobenzoic acid (PABA), as well as of Slae26 combinations with the five reference substances, 1:1 mass rate (GAE, ref.). Cell viability studies (MTS test) have revealed $IC_{50}$ values of 36 µg GAE/mL in the case of Slae26 ethanolic extract, while Slae26 combinations with the five phenolics indicated $IC_{50}$ values around 5 µg GAE/mL. In silico docking studies on the molecular targets human tankyrase 1 (TNKS1) and human tankyrase 2 (TNKS2) in complex with their native ligands, Co-crystallized 3J5A and Co-crystallized FLN, indicated score values of −104.15 and −76.97, respectively; in the series of the reference compounds studied, luteolin-7-O-glucoside was revealed with the best score values on both molecular targets (−80.49 and −85.17), together signifying real antiproliferative potential against human colon cancer of Slae26, of luteolin-7-O-glucoside, and of Slae26 combinations with all food-related bioactive compounds tested.

**Keywords:** food bioactive compounds; stokes aster ethanolic extracts; health effects; antiproliferative activity; Caco-2 cells; docking; TNKS1; TNKS2

## 1. Introduction

Green plants contain an impressive number of (phyto)compounds (estimated at several thousand distinct molecules), which are classified as either derivatives of primary metabolism or derivatives of the secondary metabolism [1–5]. Food bioactive compounds mainly belong to the secondary metabolism derivatives and they refer to compounds that have an adaptive role for green plants, as they ensure the interaction between the plant and the environment [6–9]. Among these, compounds from flavonoids subclasses (e.g., flavones, flavonols, flavanones, flavanonols, flavanols/catechins, anthocyanins, isoflavones, neoflavonoids, and chalcones) and other non-flavonoids phenolics such as phenolic acids, tannins, terpenoids, volatile oils, coumarins, lignans, and triterpene acids have been proved to have certain health benefits in humans [1,2].

In this context, food-related bioactive compounds are not only significant in their number, but also possess very complex molecules with numerous pharmacological activities and benefits for human health. The reality of benefits of plant compounds for human health support is proved by studies reporting that "about 30% of the worldwide sales of drugs

are based on natural products" [10]. Thus, it was counted that "over 60% of approved drugs are derived from natural compounds" [11], and among 185 small molecules used in the treatment of cancer, 33.5% belong to natural compounds [12]. Data have also shown that pharmaceutical companies spend approximately US $350 million to develop a new drug [13]. Furthermore, plant compounds druggability studies proved that the first four rules of Lipinski [14] do not apply to the natural products or other molecules that are the subject of an active transport system [15–17].

In terms of food-related bioactive compounds in plants, studies [18] on the common western type of food indicated that most active compounds with certain health benefits for humans come from the cocoa and tea products, berries (black chokeberry, blueberries, blackberries, red raspberries, and strawberries) and non-berries fruits (citrus, apples, black currants, plums, sweet cherries), red wine, some vegetables (artichokes, chicory, red onions, spinach), nuts (hazelnuts, pecans, almonds, walnuts), soy products, olive oil and spices (ginger, turmeric, pepper, peppermint, star anise, basil, cloves, etc.). If also considering the frequency and amount commonly used in daily diet, it was estimated that chlorogenic acid isomers and its metabolites (e.g., caffeic acid, ferulic, and isoferulic acids) and some flavan-3-ols compounds (e.g., catechins and epi(gallo)catechin gallates derivates), are the dominant food-related bioactive compounds [18].

Furthermore, the bioavailability of secondary metabolites in their natural form in plants (glycosides and esters) is generally estimated at less than 5% [19,20]. Additionally, considering the inter-individual variability in humans [21,22], it results that the benefits of food-related bioactive compounds are mostly played by the individual microbiotas, which in fact metabolize most of the ingested plant secondary metabolites.

Yet, clinical studies at the time concluded that the aglycones of plant secondary metabolites generally have the condition of good bioavailability in humans, and some glycosides and esters use some specialized enzymes from the intestine, therefore they can achieve high bioavailability in humans (over 40%); they are quercetin derivates from onions (quercetin 4′-O-beta-D-glucopyranoside namely spiraeoside) and apples (quercetin 3-O-beta-D-glucoside namely isoquercitrin), and procyanidins and epigallocatechin gallates from green tea, wine, and berries. Additionally, ferulic acid was proved to cross both, the stomach mucous layer and the blood–brain barrier, due to its combination of low molecular weight and low polarity induced to the methyl group and naringenin flavanone demonstrated the ability to influence the bioavailability of other natural and synthetic compounds [19,20]; moreover, caffeic, chlorogenic, ferulic, and ellagic acids were proved to be the subject of the active transport in humans [21–23].

Besides these, food-related bioactive compounds must pass the bio-accessibility step [24] prior to being the subject of bioavailability and absorption. Consequently, since the chemical-pharmaceutical industry can improve the efficacy of bioactive compounds from plants by the proper combination of ingredients and an appropriate formulation (e.g., by microencapsulation, use of liposomes, microemulsion into fats, etc.), the functional ingredients' food industry is disadvantaged exactly by the lack of predictability of bio-accessibility and bioavailability of phytocompounds [24,25]. However, there is clear evidence of the efficacy of plant compounds, both in the prevention and effective treatment of many human diseases [26–28]. A problem that remains to be solved is that of the active dose, even in the case of the most active plant-derived products; it involves the extracts/compounds/plant-derived products with an $IC_{50}$ smaller than 50 µg active compounds per 1 mL sample. This disadvantage of excessively high doses of plant products is even more challenging in the case of samples with certain antitumor activity and without toxic effects on the normal cells, the more so as studies show that the whole (unrefined) vegetal extracts are more active than the selective extracts or single compounds [29–32]; the higher efficacy of unrefined extracts is explained by the complexity (numerical and structural) of compounds in vegetal extracts curing the damaged tissue or overwhelming numerous defense reactions of the target.

The goal of this paper was to study the antiproliferative activity of an ethanolic extract from *Stokesia laevis* (Slae26) and its combinations with several polyphenols compounds (reference substances, ref.) on the human colon tumor cell line Caco-2 (ATCC-HTB-37), in an attempt to increase the biological activity of Slae26 [33,34] by concentrating it in own active compounds and other food-related bioactive compounds with high bioavailability and additionally benefits in cancer fight at humans. Reference compounds selected for the study were: luteolin-7-O-glucoside (cinnaroside), luteolin-8-C-glucoside (orientin), caffeic acid, gentisic acid, and p-aminobenzoic acid (PABA). Additionally, molecular docking simulation on the five phenolics (ref.) in complex with human tankyrases (TNKS1 and TNKS2), involved in colon cancer development, has been done.

## 2. Materials and Methods

**Plant material description:** The vegetal raw material (*Stokesia laevis* L.-*herba et flores*) was acquired from a specialized plant distributor in Romania, and its taxonomic identity was confirmed by the botanist's team of National Institute of Chemical-Pharmaceutical R&D, Bucharest, Romania. The fresh flowering plant was cut (the aerial part), shade dried, and ground to a medium-size powder product. Voucher specimens, the dried plant, and the powder product (Slae), are deposited in the ICCF Plant Material Storing Room.

**Chemicals, reagents and references:** Chemicals (e.g., sodium carbonate), reagents (e.g., *Folin-Ciocalteu*, *Natural Product*-NP/PEG), solvents (e.g., ethanol, ethyl acetate, formic acid, and glacial acetic acid), the reference compounds for quantitative and qualitative studies namely rutin (min. 95%), quercetin-3-O-rhamnoside (>90%), luteolin (>98%), luteolin-7-O-glucoside(>98%), luteolin-8-C-glucoside (>97%), caffeic acid (99%), chlorogenic acid (>95%), gallic acid (95%), gentisic acid (>99%), and PABA (>99%) and the cell culture reagents used, Dulbecco's Modified Essential Media (DMEM) and Foetal Bovine Serum (FBS), were purchased from Sigma-Aldrich Distributor in Romania.

**Plant extracts preparation:** Fifty (50) grams of stokes aster plant powder were extracted on the water bath (in a glass device of 2 L) with 1000 mL of 70% (*v/v*) ethanol, for one hour at the boiling temperature. The filtered ethanolic extract (Slae) in a volume of 745 mL was analyzed for qualitative (polyphenols profile) and quantitative (total phenols content) aspects. An aliquot of 250 mL of ethanol extract with the content of 0.61 mg GAE per 1 mL sample was further concentrated at sicc product (residue), and the sicc product was solved into 30.5 mL ethanol 40% (*v/v*) to achieve the content of exactly 5 mg GAE per 1 mL sample. The new (40%) ethanolic extract was sonicated 15 min at room temperature, then was filtered on the glass fiber filter. The resulting brown, homogeneous solution (Slae26) was used in pharmacological studies.

**Polyphenol's assessment in extracts:** Total phenols content was estimated using Folin–Ciocalteu reagent, by standard Romanian Pharmacopoeia (F.R.X) method [35]; results were expressed as mg gallic acid equivalents [GAE] per 1 mL sample ($R^2 = 0.9912$). Polyphenols profile in 70% stokes aster ethanolic extract was assessed by Wagner and Bladt [36] and Reich and Schibli [37] thin layer chromatography recommendation (solvent system: ethyl acetate-glacial acetic acid-formic acid-water/100:12:12:26), as described in the previous work [33].

**Pharmacological studies in vitro:** In vitro cell pharmacological studies were done on the human cancer colon cells Caco-2 (ATCC-HTB-37), using MTS test [38]. Studies were designed to evaluate both, cytotoxic activity and antiproliferative potency of singular vegetal extract Slae26, of singular polyphenol compounds (reference substances, ref.), and of Slae26 combinations with each reference substance selected for the study. The combination of the test vegetal extract (Slae26) with the five reference substances was made to provide a 1:1 mass ratio (*w/w*) between the total phenols in the plant extract (expressed as μg GAE/mL sample) and the amount of the reference compound tested, the punctual concentrations in test series of 10, 25, 50 and 100 μg total active compounds per 1 mL sample, respectively. Punctually, three-test series and corresponding control samples were prepared: (1) Slae26 test series and Slae26 solvent sample (40% ethanol) control series;

(2) polyphenol compounds test series (ref.) and their solvent sample (70% ethanol) control series; and (3) Slae26 plus reference compound test series, 1:1 mass rate (*w/w*), and their solvent sample (40% ethanol plus 70% ethanol, 1:1, *w/w*) control series. Briefly, in the MTS cytotoxicity assay, Caco-2 cells were exposed to the test/control sample series (1, 2, 3) at the time when about approximately 70% "semi-confluent" cell culture had occurred, and in the antiproliferative MTS assay, Caco-2 cells were exposed to three-test series at the time when about approximately 30% "sub-confluent" cell culture had occurred. After one or two division cell cycles (24 and/or 48 h) in the presence of test/control sample series, the culture medium is removed and Caco-2 cells are further incubated with MTS solution for another 2 h. In the end, the viability of the adherent cells is determined using CellTiter 96 AQueous One Solution Cell Proliferation Assay (Promega Corporation, Madison, WI, USA) [38]. The absorbance of the treated and control series samples is therefore measured at 490 nm with a Microplate Reader (Chameleon V Plate Reader, LKB Instruments) and the recorded values are used for cell viability estimation in vitro (see formula below). Results are presented as punctual $DO_{490}$ nm (mean $\pm$ SD, $n = 3$), and not as viability percent, to better emphasize the influence of ethanol solvent in samples.

**Molecular docking study:** Molecular docking simulations were conducted using CLC Drug Discovery Workbench (QIAGEN) in order to evaluate the reference compounds series (luteolin-7-O-glucoside, luteolin-8-C-glucoside, caffeic acid, gentisic acid and p-aminobenzoic acid) potential inhibitory activity on the human tankyrase 1 (TNKS1) and human tankarase 2 (TNKS2) enzymes, involved in cancer development in humans; the catalitic domain of human tankyrase is involved in oncogenesis, through the Wnt signaling pathway responsible for the regeneration of intestinal epithelial cells [39]. Therefore, testing new molecules with satisfactory physical–chemical properties by in silico methods against human tankyrases to block the Wnt pathway constitutes a reliable approach to hit-to-lead stage of rational drug development for intestinal cancer treatment. The PDB entree 4W6E, corresponding to human tankyrase 1 (TNKS1) in complex to the native inhibitor (encoded Co-crystallized 3J5A) [40], has been retrieved from Protein Data Bank (https://www.rcsb.org (accessed on 8 October 2021)). The tankyrase 1's preparation was realized by: removing the co-factor (Zn ion) and the water molecules; protonation; setup binding site and binding pocket at 140.29 $Å^2$. Likewise, 2-phenyl-4h-chromen-4-one, PDB entree 4HKI (encoded Co-crystallized FLN) is the native molecule inhibitor of human tankyrase 2 (TNKS2), as resulted from the systematic study on different flavonoid subclasses, reported by Narwal M. Et al. [41]; the hit compounds discovered upon their previous screening of 500 natural or naturally derived flavonoid structures, revealed that interactions within nicotinamide binding site of tankyrase 2 is responsible for inhibiting activity. The tankyrase-2's preparation was achieved by: water and other molecules (zinc ion, sulfate ion, glycerol, di(hydroxyethyl)ether) removal, protonation, ligand optimization, setup binding site and binding pocket at 121.34 $Å^2$. In both cases, the ligand's preparation was achieved by energy minimization using Spartan'14 software program from Wavefunction, Inc., Irvine, CA, USA [42]. Validation was made by re-docking the native ligands. Interactions of ligands in complex with the catalytic domain of human tankyrases were identified.

## 3. Results

### 3.1. Chemical Analytical Aspects of Ethanolic Extracts from Stokesia laevis

Slae26 is (40%) ethanolic extract standardized in total phenols content (5 mg GAE/mL) originated from the first (70%) ethanol extract (Slae) from the aerial part of *Stokesia laevis*; the first extraction in 70% ethanol is done for reasons of technological sterilization and maximum extraction of polyphenols in plant material and the second extraction (solubilization) in 40% ethanol is done for reasons of punctual extract standardization.

Polyphenol's Appraisal in Extracts
- Figure 1 (chromatograms a and b) presents the polyphenols profile of Slae and Slae26 ethanolic extracts. Therefore, HPTLC analyses of the two ethanolic extracts from

*Stokesia laevis* plant material (Slae/a and Slae26/b) indicated identical qualitative aspects, and the prevalence of two main polyphenols subclasses: caffeic acid derivates (blue fluorescent/fl. spots s3 and s5), namely chlorogenic acid (s3) and isochlorogenic acid (s5), and several luteolin derivates (yellow fl. spots s1, s2, s4, and s6); the major compound in stokes aster ethanolic extracts is in the category of luteolin monoglycosides (punctually, luteolin-7-O-glucoside/s4). Additionally, HPTLC study suggests the presence of small quantities of ellagic acid in stokes aster, punctually the blue, fl. spot at the start point of chromatograms a and b, more evident in (70%) unrefined ethanolic extract Slae than in standardized (40%) ethanolic extract Slae26.

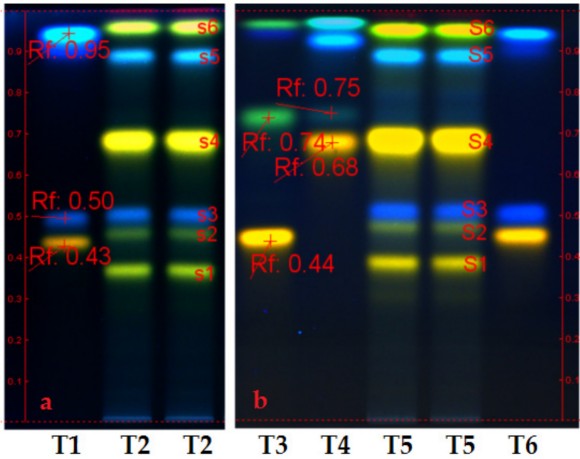

**Figure 1.** (Chromatograms **a** and **b**). Polyphenols profile in *Stokesia laevis* L. ethanolic extracts. Where: Chromatogram a). Track T1, rutin, chlorogenic acid, and caffeic acid (ref.). Tracks T2–whole (70%) ethanolic extract from *Stokesia laevis* (Slae)-duplicate sample; Chromatogram b). Track T3, rutin, vitexin, protocatechuic acid, and apigenin (ref.); Track T4, hyperoside, cosmosiin, rosmarinic acid, and kaempferol; Tracks T5–standardized (40%) ethanolic extract from Stokesia laevis (Slae26)-duplicate sample; Track T6, rutin, chlorogenic acid, and caffeic acid (ref.).

### 3.2. Pharmacological In Vitro Cytotoxicity and Antiproliferative Results

In vitro pharmacological studies were done on human colon cancer cell line Caco-2 (ATCC-HTB-37); the selection of the intestinal cells was claimed by the precondition of direct contact between the food-related bioactive compounds (ref.) and the affected tissue. Tests were done on Slae26, on the five reference compounds selected for the studies (punctually, luteolin 7-O-glucoside, luteolin 8-C-glucoside, caffeic acid, gentisic acid, and p-aminobenzoic acid), and their combinations in a manner that ensured a quantitative ratio 1:1 between the active compounds (GAE face to a particular reference compound: GAE, ref.) at identical concentrations and dilution series (punctually, 10, 25, 50 and 100 µg active compounds/mL sample); they were in parallel with a negative control sample (blank) and the solvent samples control series, 40% ethanol series for Slae26 series and 70% ethanol series for the reference compounds series, respectively. It must be reminded that the results were presented as punctual O.D.490 nm (mean $\pm$ SD, $n$ = 3), and not as current viability percent, to better emphasizes the influence of ethanol solvent in samples.

### 3.2.1. In Vitro Cytotoxicity MTS Results at 24 h and 48 h

- Cytotoxicity tests on Caco-2 cells have at the purpose to evaluate the effects of the five reference compounds (luteolin-7-O-glucoside, luteolin-8-C-glucoside, caffeic acid, gentisic acid and PABA) at punctual dilution series (10, 25, 50 and 100 µg/mL sample) comparatively to the solvent sample control series (70% ethanol series), and the negative control sample (blank), respectively. Thereby, the cytotoxicity tests also considered the evaluation of the effects of ethanol solvent in culture medium.
- The cytotoxicity results at 24 h (Figure 2) indicated that, in comparison with the negative control sample (blank), all reference compounds at concentrations smaller

than 25 µg/mL induced a stimulatory effect upon the viability of Caco-2 cells, after which they induced an inhibitory effect. The cytotoxicity results on Caco-2 cell viability at 48 h (Figure 3) showed certain inhibitory effects for all reference compounds tested and at all point series, but considering the augmented inhibitory effects of the solvent sample control series (70% ethanol), the only conclusion to be drawn is the protective effect of luteolin-7-O-glucoside, caffeic acid and gentisic acid against the cytotoxic effects of ethanol in culture medium; in the concentration interval less than 50 µg/mL sample, caffeic acid and gentisic acid only were effective against the alcohol damages in the intestinal cells.

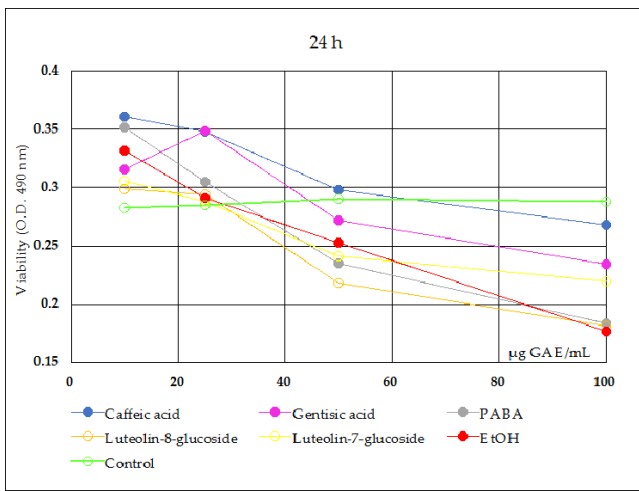

**Figure 2.** Caco-2 cell viability treated with the five reference compounds (phenolics), at 24 h.

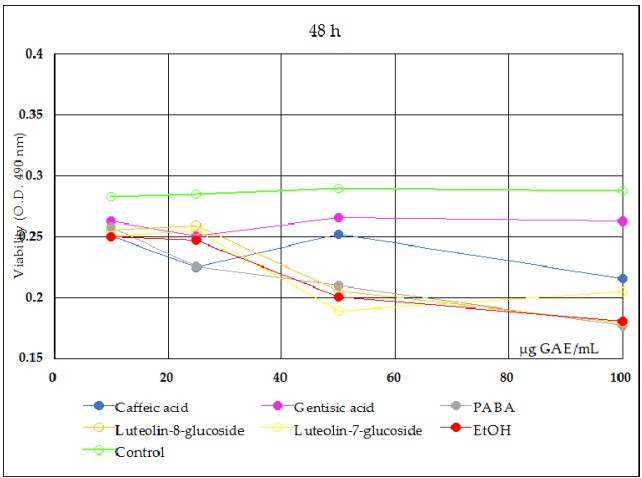

**Figure 3.** Caco-2 cell viability treated with the five reference compounds (phenolics), at 48 h.

3.2.2. In Vitro Antiproliferative MTS Test Results at 48 h

- Antiproliferative activity tests were done on Slae26, on the five polyphenols compounds, and on Slae26 combinations, 1:1 quantitative ratio (*w/w*), with the five polyphenols selected. Figure 4 presents antiproliferative activity of the five polyphenols compounds (ref.) on Caco-2 cells in comparison with the solvent sample control series (70% ethanol), 48 h after the treatment. Figure 5 presents antiproliferative activity of the test vegetal extract Slae26 on human colon cancer cell line Caco-2, and on human breast cancer cell line BT20 [33] and murine melanoma cell line B16 [34], in comparison with the solvent sample control series (40% ethanol), 48 h after the treatment. Figure 6 presents antiproliferative activity of the combinations of Slae26 with the five polyphenols compounds on human colon cancer cell line Caco-2, in comparison with

the solvent sample control series (70% ethanol and 40% ethanol, 1:1, *w/w*), 48 h after the treatment.

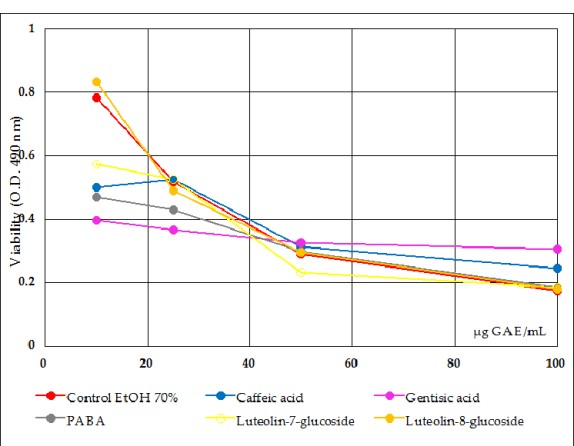

**Figure 4.** Caco-2 antiproliferative activity of the five reference compounds (phenolics), at 48 h.

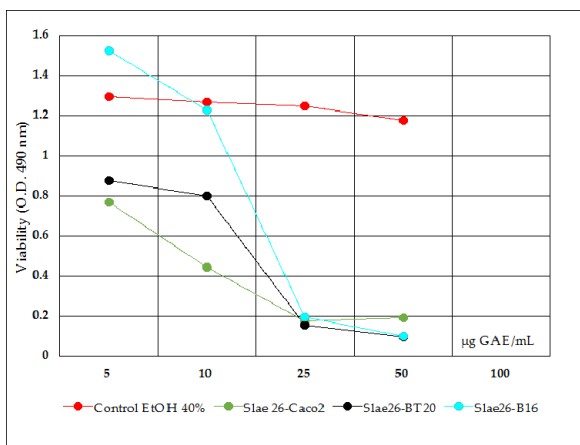

**Figure 5.** Antiproliferative activity of the Slae26, tested on Caco-2, BT20 and B16 cell lines, at 48 h.

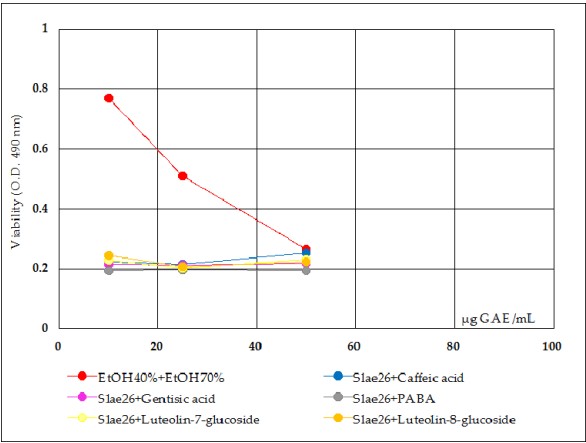

**Figure 6.** Caco-2 antiproliferative activity of the five combinations of Slae26 with the five reference compounds tested (phenolics, 1:1 *w/w* ratio of the active compounds in series), at 48 h.

- Therefore, referring to the antiproliferative potency of the five reference compounds tested (Figure 4), conclusions can be drawn only in the range of 10–25 µg/mL sample; after the threshold of 25 µg/mL sample, the effects being sensible alike to those of the control 70% ethanol sample series; at concentrations smaller than 25 µg/mL sample, gentisic acid, caffeic acid, p-aminobenzoic acid and luteolin-7-O-glucoside indicated

certain inhibitory effects upon Caco-2 cells viability, therefore an antiproliferative activity against human tumor cancer cells Caco-2.

- Slae26 test sample in comparison with 40% ethanol sample control series (Figure 5) indicated certain inhibitory activity upon the viability of human tumor cancer cell Caco-2 ($IC_{50}$ = 36 μg GAE/mL extract), similar to that previously found for human tumor breast cell BT20 ($IC_{50}$ = 42 μg GAE/mL extract) [33] and murine melanoma cell line B16 ($IC_{50}$ = 39 μg GAE/mL extract) [34].
- The antiproliferative MTS tests made on the combinations of Slae26 with the five reference compounds (Figure 6), and also in comparison with the control ethanol sample series (represented by 1:1 mixture between 40% ethanol from Slae26 sample and 70% ethanol from reference compounds samples) indicated their massive inhibitory effects upon the viability of Caco-2 cells and an $IC_{50}$ value towards 5 μg active compounds per 1 mL sample. The results on Slae26 combinations with the five plant phenolics support the possibility of boosting the antiproliferative activity of plant-derived products with $IC_{50}$ values < 50 μg active compounds/mL sample.

### 3.3. In Silico Evaluation of Reference Compounds by Docking Simulations

According to recent data [43], tankyrase 1 (TNKS1) and tankyrase 2 (TNKS2) are two homologous proteins involved in multiple pathological situations, diseases, and cancer development in humans: the two proteins (enzymes) were identified in colon cancer, gastric cancer, bladder cancer, pancreatic cancer, and also breast and brain cancer. On the other hand, human colorectal cancer is the second leading cause of cancer death worldwide [44], and statistical data have revealed that approximately 90% of worldwide human colorectal tumors show the hyper-activation of β-catenin signaling pathway, while the TNKS inhibitors were found through the regulation of the limiting factor of the β-catenin degradation protein Axin [43]. Moreover, in vitro and in vivo proof-of-concept studies established the profile of the selective inhibitors of TNKS in cancer and tankyrase linked diseases [45]. In this context, studying TNKS inhibitory potential of natural compounds is of great interest. Therefore, molecular docking simulation aimed to study potential inhibitory effects of the five phenolics in the series of food-related bioactive compounds on the molecular targets TNKS1 and TNKS2, compared to the PDB native inhibitors namely Co-crystallized 3J5A and Co-crystallized FLN.

- Results of molecular docking study (Table 1) on human tankyrase 1 (PDB ID: 4W6E) have revealed the greatest inhibitory score (−104.15) for the Co-crystallized 3J5A ligand: four hydrogen bond formations with GLU1291 (x2), SER1221 and GLY1185 amino acid residues (see Supplementary Figure S1) were counted. It is noticed that luteolin-7-O-glucoside strongly interacted with SER1221, and GLY1185 by hydrogen bonding and additionally with GLY1196 and ASP1198, achieving a docking score value of −80.49, the best among the investigated ligands. Except for Luteolin-8-C-glcoside, all molecules interact with GLY1185 and SER1221 (see Supplementary Figures S2–S6), meaning satisfactory docking scores.
- Molecular docking simulations on the human tankyrase 2 (PDB ID: 4HKI) (Table 2) have revealed luteolin-7-O-glucoside the highest inhibiting potency, as suggests its docking score (−85.17), greater than that obtained for the native PDB ligand FLN (−76.97); punctually, luteolin-7-O-glucoside interacts by eleven hydrogen bonds with the amino acids residues SER1068 and GLY1032 on chain A (see Supplementary Figure S7), and additionally, amino acid GLU1138 on chain C (see Supplementary Figure S8). Furthermore, the results place the other screened ligands in order: caffeic acid (−57.17) > gentisic acid (−51.66) > p-aminobenzoic acid (−50.42) > luteolin-8-C-glcoside (−43.47); their interactions within the active binding site of tankyrase 2 are given in Supplementary Figures S9–S12. To better identify and depict their interactions within the complexes, in Figure S13, the atomic labeling schemes for investigated structures, as arbitrarily provided by Spartan software upon energy minimization, are given.

- Table 3 depicts Lipinski's parameters for druggability assessment of the two native PDB inhibitors (3J5A and FLN), and of the five phenolics studied (ref.). The data indicate that the major difference between the two native ligands consists in the number of hydrogen bond acceptors (e.g., hydroxyl and amino -groups), HBA = 7 for 3J5 structure and HBA = 2 for FLN structure, partly explaining their significant difference in score's magnitude and inhibition effectiveness.

**Table 1.** Results of molecular docking study on human tankyrase 1 (PDB ID: 4W6E, chain A) [40].

| Ligand | Score * | RMSD | Hydrogen Bond | Length (Å) |
|---|---|---|---|---|
| Co-crystallized 3J5 (2-(4-{6-[(3S)-3,4-dimethylpiperazin-1-yl]-4-methylpyridin-3-yl}phenyl)-8-(hydroxymethyl)quinazolin-4(3H)-one) | −104.15 | 0.16 | O1 $sp^3$–O$sp^2$ GLU1291<br>O1 $sp^3$–O$sp^2$ GLU1291<br>O $sp^2$–O$sp^3$ SER1221<br>O $sp^2$–N$sp^2$ GLY1185 | 3.393<br>3.250<br>2.778<br>2.895 |
| Luteolin-7-O-glucoside | −80.49 | 0.69 | O10 $sp^3$–O$sp^2$ GLY1196<br>O10 $sp^3$–N$sp^2$ ASP1198<br>O11 $sp^3$–N$sp^2$ ASP1198<br>O11 $sp^3$–O$sp^2$ ASP1198<br>O5 $sp^3$–O$sp^2$ GLY1185<br>O3 $sp^3$–O$sp^2$ GLY1185<br>O3 $sp^3$–N$sp^2$ GLY1185<br>O3 $sp^3$–O$sp^3$ SER1221<br>O4 $sp^3$–O$sp^3$ SER1221<br>O4 $sp^3$–N$sp^2$ SER1221 | 3.284<br>2.729<br>2.741<br>3.067<br>2.456<br>3.090<br>2.647<br>3.040<br>3.087<br>3.298 |
| Luteolin-8-C-glucoside | −61.88 | 0.02 | O11 $sp^3$–N$sp^2$ ASP1198<br>O9 $sp^2$–N$sp^2$ TYR1213<br>O4 $sp^3$–N$sp^2$ HIS1201<br>O3 $sp^3$–O$sp^2$ ALA1202<br>O6 $sp^3$–N$sp^2$ HIS1201 | 2.992<br>3.034<br>2.920<br>3.129<br>2.817 |
| Caffeic acid | −58.55 | 0.15 | O3 $sp^3$–O$sp^3$ TYR1224<br>O3 $sp^3$–O$sp^2$ GLU1291<br>O2 $sp^3$–O$sp^2$ TYR1213<br>O2 $sp^3$–N$sp^2$ HIS1184<br>O2 $sp^3$–N$sp^2$ GLY1185<br>O2 $sp^3$–O$sp^2$ GLY1185 | 3.050<br>2.912<br>3.364<br>3.089<br>2.925<br>3.022 |
| Gentisic acid | −51.02 | 0.01 | O1 $sp^3$–O$sp^3$ TYR1224<br>O0 $sp^3$–O$sp^2$ TYR1213<br>O0 $sp^3$–O$sp^3$ SER1221<br>O2 $sp^3$–N$sp^2$ HIS1184<br>O2 $sp^3$–N$sp^2$ GLY1185<br>O2 $sp^3$–O$sp^2$ GLY1185 | 3.142<br>3.212<br>2.880<br>3.196<br>3.123<br>2.829 |
| p-aminobenzoic acid | −49.16 | 0.17 | N2 $sp^3$–O$sp^3$ TYR1224<br>O0 $sp^3$–O$sp^2$ TYR1213<br>O0 $sp^3$–N$sp^2$ ALA1225<br>O0 $sp^3$–O$sp^3$ SER1221<br>O0 $sp^3$–O$sp^2$ PHE1183<br>O1 $sp^2$–N$sp^2$ HIS1184<br>O1 $sp^2$–N$sp^2$ GLY1185 | 3.092<br>3.087<br>3.270<br>2.678<br>2.989<br>3.037<br>3.001 |

* The docking score used in the DDW is the PLANTS$_{PLP}$ score [46] that considers all type of contributions of heavy atom contacts (inter atom distance less than 5.5 Å) between ligand and the binding site (hydrogen bond interactions, lone-pair-metal ion interactions, non-polar interactions, non-polar-polar contacts, repulsive contacts), evaluating the potential energy change at the formation of the protein-ligand complex. Consequently, a negative score corresponds to a strong binding affinity, while a less negative or positive score means poor or lack of binding.

**Table 2.** Results of molecular docking study on human tankyrase 2 (PDB ID: 4HKI, chains A and C) [41].

| Ligand | Score | RMSD | Hydrogen Bond | Length (Å) |
|---|---|---|---|---|
| Co-crystallized FLN (2-phenyl-4h-chromen-4-one) | −76.97 | 0.02 | O4 sp$^2$–Osp$^3$ SER1068(A)<br>O4 sp$^2$–Nsp$^2$ GLY1032(A) | 2.850<br>3.029 |
| Luteolin-7-O-glucoside | −85.17 | 0.55 | O11 sp$^3$–Osp$^2$ ALA1049(A)<br>O10 sp$^3$–Osp$^2$ HIS1048(A)<br>O10 sp$^3$–Osp$^3$ SER1033(A)<br>O9 sp$^2$–Nsp$^2$ GLY1032(A)<br>O9 sp$^2$–Osp$^3$ SER1068(A)<br>O8 sp$^3$–Osp$^3$ SER1068(A)<br>O2 sp$^3$–Osp$^3$ GLU1138(C)<br>O2 sp$^3$–Nsp$^3$ LYS1067(A)<br>O6 sp$^3$–Nsp$^2$ MET1054(A)<br>O6 sp$^3$–Osp$^2$ GLY1052(A)<br>O6 sp$^3$–Osp$^3$ TYR1050(A) | 3.203<br>3.088<br>3.075<br>3.176<br>3.007<br>3.016<br>3.189<br>2.522<br>3.202<br>3.397<br>3.104 |
| Luteolin-8-C-glucoside | −43.47 | 0.67 | O11 sp$^3$–Osp$^2$ GLY1032(A)<br>O9 sp$^2$–Nsp$^2$ ILE1051(A)<br>O2 sp$^3$–Nsp$^2$ ILE1075(A)<br>O3 sp$^3$–Osp$^2$ TYR1073(A) | 2.535<br>3.075<br>2.875<br>3.256 |
| Caffeic acid | −57.17 | 0.05 | O3 sp$^3$–Osp$^2$ TYR1071(A)<br>O4 sp$^2$–Osp$^3$ GLU1138(C)<br>O1 sp$^3$–Nsp$^2$ ALA1062(A)<br>O1 sp$^3$–Osp$^2$ PHE1030(A)<br>O1 sp$^3$–Osp$^3$ SER1068(A)<br>O2 sp$^3$–Nsp$^2$ HIS1031(A)<br>O2 sp$^3$–Nsp$^2$ GLY1032(A)<br>O2 sp$^3$–Osp$^2$ GLY1032(A) | 3.003<br>2.651<br>3.217<br>3.016<br>2.397<br>2.988<br>2.636<br>3.005 |
| Gentisic acid | −51.66 | 0.02 | O1 sp$^3$–Osp$^3$ TYR1071(A)<br>O2 sp$^3$–Osp$^2$ GLY1032(A)<br>O2 sp$^3$–Nsp$^2$ GLY1032(A)<br>O2 sp$^3$–Nsp$^2$ HIS1031(A)<br>O0 sp$^3$–Osp$^3$ SER1068(A)<br>O0 sp$^3$–Osp$^2$ PHE1030(A)<br>O0 sp$^3$–Osp$^2$ TYR1060(A) | 3.118<br>2.873<br>3.044<br>3.158<br>2.800<br>3.258<br>3.1817 |
| p-aminobenzoic acid | −50.42 | 0.16 | O0 sp$^3$–Osp$^3$ GLU1138(C)<br>O1 sp$^2$–Osp$^3$ TYR1071(A)<br>N2 sp$^3$–Nsp$^2$ HIS1031(A)<br>N2 sp$^3$–Nsp$^2$ GLY1032(A)<br>N2 sp$^3$–Osp$^2$ GLY1032(A) | 2.848<br>3.048<br>2.987<br>2.650<br>2.829 |

**Table 3.** Lipinski's parameters for druggability assessment.

| Ligand | HBD | HBA | LogP | LV | rb |
|---|---|---|---|---|---|
| Co-crystallized 3J5A (PDB ID: 4W6E) | 1 | 7 | 4.66 | 0 | 4 |
| Co-crystallized FLN (PDB ID: 4HKI) | 0 | 2 | 4.80 | 0 | 1 |
| Luteolin-7-O-glucoside | 7 | 11 | 1.57 | 2 | 4 |
| Luteolin-8-C-glucoside | 8 | 11 | −0.70 | 2 | 3 |
| Caffeic acid | 3 | 4 | 1.58 | 0 | 2 |
| Gentisic acid | 3 | 4 | 1.49 | 0 | 1 |
| p-aminobenzoic acid | 3 | 3 | 0.97 | 0 | 1 |

Where: HBD–hydrogen bond donors; HBA–hydrogen bond acceptors; logP–water-octanol partition coefficient; LV–Lipinski's violations; rb–rotatable bonds count.

## 4. Discussion

The objectives proposed in the present work were to evaluate the antiproliferative activity on human colon cancer cell Caco-2 of the ethanolic extract from *Stokesia laevis* plant species (codified Slae26), of five food-related polyphenols compounds (reference substances, ref.) selected based on precise criteria, and of Slae26 combinations with the five reference substances in a manner that ensured 1:1 mass rate (*w/w*) between the active compounds in the combined samples (GAE, ref.). The selection of the five polyphenols compounds, namely luteolin-7-O-glucoside (cinnaroside), luteolin-8-C-glucoside (orientin), caffeic acid, gentisic acid, and p-amino benzoic acid (PABA), is supported by the following criteria: Luteolin monoglycosides are the major flavonoid compounds in *Stokesia laevis* ethanolic extracts, and the main contributor on antiproliferative activity of Slae26 in the authors' works [33,34], as also results from the literature data today [47–51]; caffeic acid is the foremost compound in human diet resulting from the metabolism of (iso)chlorogenic acid esters, also the subject of active transport in humans, therefore it can be considered the food-related bioactive compound with the largest and constant presence in the human intestine; moreover, studies proved its ability to protect the human peripheral blood lymphocytes against the cellular damages by gamma radiation (gamma radiation is used in radio-oncology to treat cancer patients), therefore the combination of a natural antiproliferative product with a natural radiation guardian is of interest for humans [52]; gentisic acid is a highly active antioxidant compound proven to enhance the activity of antioxidant enzymes in humans, at the same time acting as an inhibitor of lipid peroxidation in cell membranes, and is also demonstrated to effectively protect human erythrocytes against gamma radiation exposure [53–55]; benzoic acid is a natural phenolic acid with antibacterial activity [56] used as a preservative in food [57]; it also can be found in fermentation products such as Caciocavallo and mozzarella cheeses [58].

In this context, cytotoxicity MTS tests were designed to investigate both the effects of the five reference compounds applied on Caco-2 cells at the time when about 70% of the predicted "sub-confluent" cells had occurred, and the effects on Caco-2 cells viability of ethanol solvent in the test samples. Strong inhibitory effects of control (70%) ethanol solvent series on the viability of Caco-2 cells, more augmented than that of the test samples, made it impossible to draw a conclusion on the cytotoxicity of the reference compounds; yet, caffeic and gentisic acid diagrams' aspect at 24 h and 48 h both suggest their ability to protect intestinal cells against the cytotoxic effects of ethanol solvent.

Antiproliferative MTS tests on the five reference compounds, applied on Caco-2 cells at the time when about 30% of the predicted "sub-confluent" cells had occurred, indicated certain antiproliferative effects of gentisic acid, caffeic acid, p-aminobenzoic acid (PABA) and luteolin-7-O-glucoside in the range of 10–25 µg/mL sample, after that the activity of the reference compounds series being substantially matching with that of the control solvent (70% ethanol) dilution series. The Slae26 test sample, in comparison with own control solvent (40% ethanol) dilution series, indicated certain inhibitory activity of the vegetal extract upon the viability of the human tumor cancer cell Caco-2 ($IC_{50}$ = 36 µg GAE/mL sample), and therefore is similar to that previously found for human breast cancer cell line BT20 ($IC_{50}$ = 42 µg GAE/mL sample) [33] and murine melanoma cell line B16 ($IC_{50}$ = 39 µg GAE/mL sample) [34]. Slae26 combinations with the five reference compounds (1:1 ratio between the active compounds in samples) in comparison with the control ethanol sample series (1:1 ratio between 40% ethanol solvent in Slae26 sample and 70% ethanol solvent in reference samples) indicated significant increases of inhibitory effects upon the viability of Caco-2 cells ($IC_{50}$ ~ 5 µg/mL sample), without interfering toxic effects of the ethanol solvent in samples. Punctually, the $IC_{50}$ value of Skae26 was reduced from a 36 µg/mL sample to about a 5 µg/mL sample, signifying real therapeutic usefulness and antitumor potency on human colon cancer cell of Sla26 combinations with the five food-related bioactive compounds.

Docking studies on the molecular target TNKS1 indicated the greatest inhibitory activity (score value: −104.15) for the native ligand, co-crystallized 3J5A, which established

interactions (hydrogen bonds) with three amino acid residues in the active binding site of chain A (SER1221, GLY1185, and GLY1196). In the series of test polyphenol compounds, luteolin-7-O-glucoside showed the best docking score on TNKS1 (score value: $-80.49$) and established interactions with two of three amino acids (SER1221, GLY1185) in the binding site. The other scores were: luteolin-8-C-glucoside ($-68.88$) > caffeic acid ($-58.55$) > gentisic acid ($-51.02$) > p-aminobenzoic acid ($-49.16$). In the case of molecular target TNKS2, luteolin-7-O-glucoside emphasized the best docking score ($-85.17$), greater than that of the native ligand, Co-cristallized FLN ($-76.97$); luteolin-7-O-glucoside interacted with two amino acids residues (SER1068 and GLY1032) on chain A, and one amino acid residue (GLU1138) on chain C. The other scores were: caffeic acid ($-57.17$) > gentisic acid ($-51.66$) > p-aminobenzoic acid ($-50.42$) > luteolin-8-C-glcoside ($-43.47$). Overall, docking studies on the human tankyrase enzymes have confirmed antiproliferative potency of bioactive compounds from food products on human colon cancer cells, especially of luteolin-7-O-glucoside.

Other food-related bioactive compounds that have been proven to have antiproliferative activity against lung, colon, breast, and prostate cancer cells are terpenoids and volatile oils (EOs) from *Lamiaceae* species [59]. Proving these, in vitro pharmacological studies on the volatile oils (EOs) extracted from the aerial part of lavender species (*Lavandula hybrid* Rev., *Lavandula latifolia* Medikus., *Lavandula vera* D.C. and *Lavandula angustifolia* Miller cultivated in Bulgaria), MTS test on Caco-2 cells and Caco-2 cell morphology analysis by electron microscopy before and after the treatment with EOs, respectively, both experiments have been confirmed to have augmented cytotoxic effects of EOs on the colon cancer, and at the same time better effectiveness of EOs in comparison with the separate constituents (e.g., linalool and 1,8-cineole), explained by their synergistic and more selective effect [60]. There were also revealed antimicrobial and antioxidant effects of EOs from lavender species, which are very usefully in managing intestinal diseases in humans [60,61].

Furthermore, studying the transport of caffeic and chlorogenic acids in Caco-2 cells, Konishi et al. [62] noticed a greater efficiency of absorption of caffeic acid, and the fact that polarized transport of caffeic acid has been inhibited by benzoic acid and acetic acid, two representative substrates of MCT (monocarboxylic acid transporter) in the intestine. They also revealed that the intestinal transport took place mainly by paracellular diffusion, and that m-coumaric acid and 3-(m-hydroxyphenyl) propionic acid, the main metabolites of chlorogenic and caffeic acid through the colonic microflora, competitively inhibited the transport of fluorescein, also known as a substrate of MCT. Together, these results proved the physiological importance of MCT-mediated absorption, both in the case of phenolic acids themselves and their colonic metabolites. Regarding the potential health benefits by combining flavonoids with other bioactive compounds in plants, aiming to investigate the antioxidant activity of the combination of luteolin and chlorogenic acid, Hsieh et al. studies [63] indicated that the half-maximal inhibitory concentration of luteolin and chlorogenic (IC$_{50}$ of luteolin was estimated at about 26 μg/mL and of chlorogenic acid at about 86 μg/mL) combined in a 1:1 ratio resulted in an augmented increase of free radical oxygen inhibition (78%), compared to that of the separate compounds. Finally, in vivo studies [64] on the diabetic rats with gastroenteropathy induced by NSAIDS (Diclofenac sodium/DCF; 7.5 mg/kg, PO, b.i.d.) indicated that kaempferol, ascorbic acid, lupeol, diosgenin, β-sitosterol, stigmasterol, and β-amyrin acted synergistically in combination with *Costus pictus* ethanolic extract (CP); the punctual combination of quercetin (Q) with *Costus pictus* ethanolic extract (QCT; 50 mg/kg for total 10 days) indicating it would be a promising candidate for treating patient with type 2 diabetes mellitus and NSAID induced-gastroenteropathy.

## 5. Conclusions

Further studies to assess the evolution of antiproliferative activity of Slae26 combinations with polyphenols compounds upon the tumor cell lines Caco-2, BT20 and B16, at lower concentrations and on several consecutive days, are proposed.

**Supplementary Materials:** The following are available online at https://www.mdpi.com/article/10.3390/app11219944/s1, Figure S1: Hydrogen bond interactions of co-crystallized 3J5 with TNKS1 (4W6E), Figure S2: Hydrogen bond interactions of luteolin-7-O-glcoside with TNKS1 (4W6E), Figure S3: Hydrogen bond interactions of luteolin-8-C-glcoside with TNKS1 (4W6E), Figure S4: Hydrogen bond interactions of caffeic acid with TNKS1 (4W6E), Figure S5: Hydrogen bond interactions of gentisic acid with TNKS1(4W6E), Figure S6: Hydrogen bond interactions of p-aminobenzoic acid with TNKS1(4W6E), Figure S7: Hydrogen bond interactions of Co-crystallized FLN with TNKS2 (4HKI), Figure S8: Hydrogen bond interactions of luteolin-7-O-glcoside with TNKS2 (4HKI), Figure S9: Hydrogen bond interactions of luteolin-8-C-glcoside with TNKS2 (4HKI), Figure S10: Hydrogen bond interactions of caffeic acid with TNKS2 (4HKI), Figure S11: Hydrogen bond interactions of gentisic acid with TNKS2 (4HKI), Figure S12: Hydrogen bond interactions of p-aminobenzoic acid with TNKS2 (4HKI), Figure S13: Atomic labeling scheme for A: luteolin-7-O-glucoside; B: luteolin-8-C-glucoside; C: caffeic acid; D: gentisic acid; E: aminobenzoic acid, as arbitrary provided by Spartan software upon energy minimization.

**Author Contributions:** Conceptualization, L.C.P.; methodology, G.N. and A.A.; formal analysis, A.S. and L.P.; investigation. G.N., A.A. and L.C.P.; resources, L.C.P.; writing—original draft preparation, L.C.P.; writing—review and editing, L.C.P. and A.S.; visualization, A.S.; supervision, L.C.P.; project administration, L.C.P.; funding acquisition, L.C.P. All authors have read and agreed to the published version of the manuscript.

**Funding:** This paper has been financed through the Program NUCLEU, which is implemented with the ANCSI support, project no. PN 19-41 01 02.

**Conflicts of Interest:** The authors declare no conflict of interest.

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
