# Peer review of "Antiproliferative Activity of Stokesia laevis Ethanolic Extract in Combination with Several Food-Related Bioactive Compounds; In Vitro (Caco-2) and In Silico Docking (TNKS1 and TNKS2) Studies"

_applsci, doi:10.3390/app11219944_

Round 1

Reviewer 1 Report

The article deals with the use of reference molecules with bioactive properties as additives to ethanolic extracts of Stokesia laevis for promoting the antiproliferative activity on human colon cancer cells. This article results to be interesting, pertinent, and innovative. In general, the article is well written, but lacks a detailed revision or supervision of the final document. Some punctuation and redaction details are highlighted. Finally, the Abstract is confusing, lacking clarity. It must be rewritten.

Line 14.           Check the spaces in “GAE/mL”.

Line 18.           Change “GAE : ref.” for “GAE, ref.”.

Line 19.           Change “sample), of the reference…” for “sample) of the reference…”.

Line 24.           Change “tested, and revealed” for “tested and revealed”.

Check the abstract redaction.

Line 37.           Change “conclusion of a greater flexibility” for “conclusion of greater flexibility”.

Line 44.           Change “in basic” for “in the basic”.

Line 45.           Change “differentiate distributed” for “differentially distributed”.

Line 59.           Change “number, but” for “number but”.

Line 59.           Change “complexity being due” for “complexity is due”.

Line 60.           Change “Data also shown” for “Data have also shown”.

Line 79            Change “plant compounds druggability” for “plant compounds’ druggability”.

Line 85.           Add a comma after raspberries.

Line 90.           Add a comma after ferulic.

Line 105.         Change “from intestine” for “from the intestine”.

Line 106.         Change “in humas” for “in humans”.

Line 109.         Add a comma after wine.

Line 113.         Add a comma after ferulic.

Line 121.         Change “of food matrix” for “of the food matrix”.

Line 123.         Change “chemical pharmaceutical” for “chemical-pharmaceutical”.

Line 126.         Change “functional ingredients food industry” for “functional ingredients’ food industry”.

Line 131          Change “plant derived” for “plant-derived”.

Line 132          Change “plant derived” for “plant-derived”.

Line 134.         Change “is even more challenges” for “is even more challenging”.

Line 139.         Change “of target tissue” for “of the target tissue”.

Line 147.         Add a comma after gentisic acid.

Line 149.         Change “have been done” for “has been done”.

Line 155.         Add a comma after shade dried.

Line 158.         Add a comma after reagents.

Line 160.         Add a comma after formic acid.

Line 162.         Add a space before caffeic acid.

Line 192.         Change “three test” for “three-test”.

Line 203.         Change “At the end” for “In the end”.

Line 213.         Change “luteolin-7-O-glcoside” for “luteolin-7-O-glucoside”.

Line 213.         Change “luteolin-8-C-glcoside” for “luteolin-8-C-glucoside”.

Line 216.         Change “for regeneration” for “for the regeneration”.

Line 219.         Change “hit-to-led” for “a hit-to-lead”.

Line 221.         Change “retrieved form Protein Data Bank” for “retrieved from Protein Data Bank”.

Line 223.         Add a comma after binding site.

Line 225.         Change “U.S.A” for “U.S.A.” or “USA”.

Line 236.         Change “(chromatogram a and chromatogram b)” for “(chromatograms a and b)”.

Line 236.         Change “presents polyphenols profile” for “presents the polyphenols profile”.

Line 241.         Add a comma after s4.

Line 255.         Change “GAE face to particular” for “GAE face to a particular”.

Line 265.         Add a comma after gentisic acid.

Line 272.         Add a comma after that.

Line 278.         Change “1 mL sample)” for “1 mL sample”.

Line 285.         Change “with to the solvent” for “with the solvent”.

Line 298.         Change “para aminobenzoic acid” for “p-aminobenzoic acid”.

Line 316.         Add a comma after diseases.

Line 319.         Change “the human” for “human”.

Line 320.         Change “statistic data” for “statistical data”.

Line 333.         Add a comma after SER1221.

Line 337.         Change “Except Luteolin-8-C-glcoside” for “Except for Luteolin-8-C-glucoside”.

Line 342.         Change “Chromatogram a.” for “Chromatogram a)”.

Line 342.         Add a comma after chlorogenic acid.

Line 343.         Change “Chromatogram b.” for “Chromatogram b)”.

Line 344.         Add a comma after protocatechuic acid.

Line 345.         Add a comma after rosmarinic acid.

Line 346.         Add a comma after chlorogenic acid.

Line 349.         Add a period at the end of the phrase.

Line 351.         Add a period at the end of the phrase.

Line 353.         Add a period at the end of the phrase.

Line 355.         Add a period at the end of the phrase.

Line 357.         Add a period at the end of the phrase.

Table 1.           Change “Luteolin-7-O-glcoside” for “Luteolin-7-O-glucoside”.

Table 1.           Change “Luteolin-8-C-glcoside” for “Luteolin-8-C-glucoside”.

Table 2.           Change “Luteolin-7-O-glcoside” for “Luteolin-7-O-glucoside”.

Table 2.           Change “Luteolin-8-C-glcoside” for “Luteolin-8-C-glucoside”.

Line 373.         Add a comma after gentisic acid.

Line 373.         Change “para-amino benzoic acid” for “p-amino benzoic acid”.

Line 400.         Change “para-amino benzoic acid” for “p-amino benzoic acid”.

Line 423.         Change “luteolin-8-C-glcoside” for “luteolin-8-C-glucoside”.

Line 424.         Change “para-amino benzoic acid” for “p-amino benzoic acid”.

Line 428.         Change “Kobayashi et al” for “Kobayashi et al.”.

Line 436.         Change “investigate antioxidant” for “investigate the antioxidant”.

Line 436.         Change “combination of luteolin” for “the combination of luteolin”.

Line 437.         Change “Hsieh et al” for “Hsieh et al.”.

Line 438.         Change “half maximal” for “half-maximal”.

Line 439.         Change “26 µg·/mL” for “26 µg/mL”.

Line 442.         Change “gastroenteropaty” for “gastroenteropathy”.

Line 444.         Add a comma after stigmasterol.

Line 448.         Change “gastroenteropaty” for “gastroenteropathy”.

Reference 38 is wrongly reported.

The Digital Object Identifiers of all the references must be included.

Author Response

Esteemed Reviewer,

First of all, thank you very much for taking the time to assess our manuscript.   Based on your very careful review of our paper, your comments and suggestions, the manuscript has been significantly improved.  

Therefore, the authors REVISED the English grammar, and the punctual errors signaled in Lines 14 - 448; the authors revised the title, rebuilt the abstract, shortened the introduction, and completed with docking studies on the human tankyrase 2 [41]; the discussion on the docking results has been deepen, and there were added literature data on the antiproliferative activity of volatile oils [59-61]; the authors revised the references, and added the Digital Object Identifiers.   

We hope that the revised paper answers to the punctual and general comments of the reviewers and you will accept the publication of the manuscript entitled “Antiproliferative activity of Stokesia laevis ethanolic extract in combination with several food-related bioactive compounds; in vitro (Caco-2) and in silico docking (TNKS1 and TNKS2) studies” .

Sincerely yours, Lucia Pirvu et al.

Reviewer 2 Report

  1. Introduction is unnecessarily long. Authors are advised to focus on the importance of Stokesia laevis and the polyphenol compounds used in the study and rewrite the introduction.
  2. Is there any relation between the docking score and inhibitory activity? Please prove it with suitable references. The discussion related to the molecular docking should also be elaborated.
  3. Additionally, docking of tankyrase 2 is also recommended. 

Author Response

Esteemed Reviewer,

First of all, thank you very much for taking the time to assess our manuscript.    Based on your very careful review of our paper, your comments and suggestions, the manuscript has been significantly improved.   

Therefore, the authors REVISED the English grammar, and the punctual errors signaled in Lines 14 - 448; the authors revised the title, rebuilt the abstract, shortened the introduction, and completed with docking studies on the human tankyrase 2 [41]; the discussion on the docking results has been deepen, and there were added literature data on the antiproliferative activity of volatile oils [59-61]; the authors revised the references, and added the Digital Object Identifiers.  We also mention that the docking score used in the Drug Discovery Workbench is the PLANTSPLP score [46] that considers all type of contributions of heavy atom contacts (inter atom distance less than 5.5 Å) between ligand and the binding site (hydrogen bond interactions, lone-pair-metal ion interactions, non-polar interactions, non-polar-polar contacts, repulsive contacts), evaluating the potential energy change at the formation of the protein-ligand complex. Consequently, negative score corresponds to a strong binding affinity. At the contrary, less negative or positive score mean poor or lack of binding.

We hope that the revised paper answers to the punctual and general comments of the reviewers and you will accept the publication of the manuscript entitled “Antiproliferative activity of Stokesia laevis ethanolic extract in combination with several food-related bioactive compounds; in vitro (Caco-2) and in silico docking (TNKS1 and TNKS2) studies.

Sincerely yours, Lucia Pirvu et al.

Reviewer 3 Report

Dear authors 

1 - Introduction: must be reformed in the content and in the writing of the general part review the syntax of the topic

2- Discussion : deepen (max two lines) the use of essential oils and the mechanism against cancer cells such as CaCo2, and their use in the future. Further information on this using and citing the following references: PMID: 28114805 ; PMID: 33066157 ; PMID: 32982183

3 - Check the bibliographic entries throughout the text, some of which are non-compliant, review some entries in the references and necessarily insert those referred to in point 2 for the purpose of acceptance by me.

4 - Review the English grammar and in particular the applied scientific English: in particular, verbal tenses and syntax in the discussion

Author Response

Esteemed Reviewer,

First of all, thank you very much for taking the time to assess our manuscript.   Based on your very careful review of our paper, your comments and suggestions, the manuscript has been significantly improved.   

Therefore, the authors REVISED the English grammar, and the punctual errors signaled in Lines 14 - 448; the authors revised the title, rebuilt the abstract, shortened the introduction, and completed with docking studies on the human tankyrase 2 [41]; the discussion on the docking results has been deepen, and there were added literature data on the antiproliferative activity of volatile oils [59-61]; the authors revised the references, and added the Digital Object Identifiers.   

We hope that the revised paper answers to the punctual and general comments of the reviewers and you will accept the publication of the manuscript entitled “Antiproliferative activity of Stokesia laevis ethanolic extract in combination with several food-related bioactive compounds; in vitro (Caco-2) and in silico docking (TNKS1 and TNKS2) studies”.

Sincerely yours, Lucia Pirvu et al.

Round 2

Reviewer 2 Report

The paper is acceptable in the present form.